# Anthroponotic-Based Transfer of *Staphylococcus* to Dog: A Case Study

**DOI:** 10.3390/pathogens11070802

**Published:** 2022-07-15

**Authors:** Massimiliano Orsini, Sara Petrin, Michela Corrò, Giulia Baggio, Elena Spagnolo, Carmen Losasso

**Affiliations:** 1Laboratory of Microbial Ecology and Genomics, Istituto Zooprofilattico Sperimentale delle Venezie, Viale dell’Università 10, 35020 Legnaro, Italy; spetrin@izsvenezie.it (S.P.); gbaggio@izsvenezie.it (G.B.); 2Department of Diagnostics in Animal Health, Istituto Zooprofilattico Sperimentale delle Venezie, Viale dell’Università 10, 35020 Legnaro, Italy; mcorro@izsvenezie.it (M.C.); espagnolo@izsvenezie.it (E.S.)

**Keywords:** *Staphylococcus aureus*, anthroponosis, amphyxenosis

## Abstract

Although usually harmless, *Staphylococcus* spp. can cause nosocomial and community-onset skin and soft tissue infections in both humans and animals; thus, it is considered a significant burden for healthcare systems worldwide. Companion animals have been identified as potential reservoirs of pathogenic *Staphylococcus* with specific reference to Methicillin Resistant *Staphylococcus aureus* (MRSA). In this study, we investigated the circulation and the genetic relationships of a collection of *Staphylococcus* spp. isolates in a family composed of four adults (a mother, father, grandmother, and grandfather), one child, and a dog, which were sampled over three years. The routes of transmission among humans and between humans and the dog werelyzed. The results displayed the circulation of many *Staphylococcus* lineages, belonging to different species and sequence types (ST) and being related to both human and pet origins. However, among the observed host-switch events, one of them clearly underpinnthroponotic route from a human to a dog. This suggests that companion animals can potentially have a role as a carrier of *Staphylococcus*, thus posing a serious concern about MRSA spreading within human and animal microbial communities.

## 1. Introduction

*Staphylococcus aureus* (*S. aureus*) is a common bacterium present on the skin and mucous membranes in 20–30% of human healthy subjects [1]. Although usually harmless, it can be the cause of nosocomial and community-onset skin and soft tissue infections both in humans and animals; thus, it is considered a significant burden for healthcare systems worldwide [2].

Some strains of *S. aureus* can develop resistance to beta-lactam antibiotics such as penicillin, which are widely used to treat human infections [3]. These strains are known as methicillin-resistant *Staphylococcus aureus* (MRSA).

Humans become infected by *S. aureus* mainly through direct contact with infected persons or with medical instruments and equipment. In recent years, companion animals, such as dogs, cats, and horses, have been identified as potential reservoirs of *S. aureus* and specifically of MRSA [4]. This assumption derives from many case reports that have documented *S. aureus* infections in animal owners, associated with colonization through genetically related strains in their pets [5]. On the other hand, there are suggestions that *S. aureus* carriage is not sustained for long periods by companion animal hosts in a clean environment [4].

It is a matter of fact that *S. aureus* exhibits tropisms to many distinct animal hosts [6]. The infection is often maintained in both humans and lower vertebrate animals and transmitted in either direction (amphixenoses). Host switching is not infrequent; it is often accompanied by specific genetic signatures such as a loss of phages, which are known to play a role in human colonization [7]. The anthroponotic transmission is not infrequent either [8], and can be the cause of the human-to-human spread of potentially virulent strains via animal-supported routes [9]. In addition to those already mentioned routes of infection, a third hypothesis is plausible, and it consists of the possibility of an anthroponotic-based transfer of *S. aureus* and MRSA to companion animals. In this case, anthroponosis refers to a human infectious disease that can be naturally transmitted to other animals, and it is considered the reverse of the zoonosis, where a pathogen or a parasite infects primarily an animal but can also infect and cause disease in humans. Thus, based on the existing evidence, whether companion animals are true reservoirs of *S. aureus* and in particular MRSA or whether they should only be considered as contaminated living vectors is a topic that is still debated and is worthy of further study.

In this study, we tested this hypothesis by investigating the circulation and the genetic relationships of isolates of *Staphylococcus* spp. in a family composed of four adults (a mother, father, grandmother, and grandfather), one child, and a dog, which were sampled because of recurrent or relapsing folliculitis, which was resolved through general and local therapy. The routes of transmission among humans and between humans and the dog were also estimated.

## 2. Results

### 2.1. Clinical Context

A single family, composed of a father, mother, grandfather, grandmother, a child, and a dog, was enrolled and regularly sampled in the period 2017–2019. Three adults suffered from recurrent folliculitis: the father had recurrent folliculitis affecting several body districts since 2017, and he was treated with cephalosporin (400 mg die for ten days through oral administration) and chlorhexidine (1% topic daily administration) and then the infection was resolved in 2018; the grandfather suffered from scrotal folliculitis in 2018, which required surgery, and a recurrent skin folliculitis in the same year, which was successfully treated in the same way; and the grandmother had purulent folliculitis in various body areas in 2018, which resolved in 2019 as well. The mother and the child were asymptomatic.

### 2.2. Dataset Description

A total of 25 isolates were collected in the present study (Table 1). Seven samples were collected from the dog and were identified as *S. pseudintermedius* (*n* = 4), *S. aureus* (*n* = 2), and *S. capitis* (*n* = 1). The remaining isolates were identified as *S. aureus* (*n* = 13), *S. epidermidis* (*n* = 3), *S. lugdunensis* (*n* = 1), and *S. warneri* (*n* = 1), and were collected from different subjects from the same family over the three-year period considered.

### 2.3. Genomic Analysis

The draft assemblies ranged in size from 2.46 to 2.92 Mb with a range of 2312 to 3276 protein coding sequences (full assembly metrics are described in Appendix A). The 7-loci MLST analysis returned a reliable ST for all *S. aureus* isolates, which belonged to five different STs: ST8 was identified in 2017; ST8, ST10, ST22, ST59, ST7204, and ST707 were identified in 2018; and ST10, ST22, and ST59 were identified in 2019 (Table 1). The *S. epidermidis* strains showed a more heterogeneous profile, with a strain belonging to ST5, one belonging to ST153, and one belonging to ST1133 (Table 1, Appendix A). Interestingly, two novel *S. pseudintermedius* STs (ST2168 and ST2169) and one new *S. aureus* ST (ST7204) were identified and submitted to the pubMLST database [https://pubmlst.org accessed on 15 January 2022]. All these new STs were isolated from the dog in 2017, 2018, and 2019. A MLST scheme is not available for *S. warnerii* and *S. capitis*.

### 2.4. Antibiotic Resistance Profiling

The results of the presence of antibiotic-resistance genes showed a generalized resistance to the beta-lactam antibiotics, encoded by the *blaZ* gene, with the exclusion of the *S. warnerii* and the *S. lugdunensis* strains (Table 2). Two profiles of genes carrying the resistance to aminoglycoside antibiotics have been observed. In the first, shared by *S. pseudointermedius* and *S. aureus* ST8 strains, the resistance is encoded by the *ant(6)-ia* and the *aph(3)-III* genes; a second pattern consisting of *aadD*, *aac(6)-aph(2)*, and *cat(pC221)* genes was observed in *S. epidermidis* strains. The presence of the *cat(pC221)* gene was also observed in the two *S. pseudointermedius* ST63 strains. The Fosfomycin resistance gene *fosB* was only detected in the three *S. epidermidis* strains. The *mecA* gene, conferring resistance to methicillin, was found in all the *S. aureus* ST8 strains, including the strain isolated from the dog; moreover, it was found in an *S. epidermidis* ST5 strain (Sep_19_GM_ST5_150797). The two genes *mphC* and *msrA*, conferring resistance to macrolides and to erythromycin and streptogramin B, respectively, were found in *S. aureus* ST8 and one ST10 strain isolated in 2019, in addition to the *S. epidermidis* Sep_17_CH_ST1133_123222 strain. One erythromycin inducible gene (*ermB*) conferring resistance to streptogramin and macrolides was found in all the *S. pseudointermedius* strains, while the *ermC* gene was found in one *S. epidermidis* strain (Sep_19_GM_ST5_150797). Finally, the *S. epidermidis* Sep_19_GM_ST5_150797 strain showed the presence of two genes (*vgaA* and *vgaALC*) conferring resistance to streptogramin A antibiotics and related compounds.

### 2.5. Prophages Analysis

A total of 10 different intact prophages were identified in the whole dataset (Table 3, extended results are described in Appendix A). The analysis revealed that prophages showed a strictly species-specific and ST-specific pattern, with the richest clusters being *S. aureus* ST10 and *S. aureus* ST8, with 4 and 3 different prophages detected, respectively (Table 3). The *S. capitis* Sca_19_DO_STX_150837 sample showed the presence of a single, intact prophage (Staphy_StB12_NC_020490) that was not observed in any other sample. A different pattern was observed for the two *S. aureus* samples isolated from the dog (Sau_18_DO_ST7204_153686-1 and Sau_17_DO_ST8_123220), belonging to ST7204 and ST8, respectively. Interestingly, the former, despite differing from the sample isolated from the child in the same period (Sau_18_CH_ST707_153686-4) by only one allele (*arcC*), shared the same prophage pattern with it.

### 2.6. Plasmids Content

A total of 13 plasmid replicons were observed (Table 4), and for the majority of them it was possible to identify a pattern of presence linked to the sequence type. For example, the *rep19*, *rep7c*, and the *rep16* plasmid replicons were observed in the *S. aureus* ST8 samples only, while the *rep16* plasmid replicon was further observed in the *S. aureus* ST707 and ST7204 samples, together with the *repUS5* plasmid replicon as well, and shared with the Sau_19_MO_ST10_150824 strain. The *S. epidermidis* strain Sep_19_GM_ST5_150797 displayed a unique pattern consisting of *rep7a*, *repUS19*, *repUS12*, and *rep5b*. Interestingly, the *rep7a* plasmid replicon was observed in samples of different species; namely, it was present in *S. epidermidis* (3/3), *S. lugdunensis* (1/1), *S. pseudintermedius* (2/4), and *S. aureus* ST8 (1/5) isolates. On average, when present, one to three plasmid replicons were identified in the samples, and only one sample (Sep_19_GM_ST5_150797) carried four plasmid replicons. Conversely, samples belonging to *S. aureus* ST22 and ST59, in addition to two *S. aureus* ST10 isolates, did not show any contig-carrying plasmid replicons. Some other plasmid replicons, such as *rep20*, *rep39*, *repUS35*, and *rep22*, were sporadically observed in Sca_19_DO_STX_150837, Sep_17_CH_ST1133_123222, Swa_19_FA_STX_150815, and Sep_17_CH_ST1133_123222, respectively.

### 2.7. IEC Genes

The results of the IEC gene sequence search at the protein level are shown in Table 5. The *scn* gene was found in all samples, highlighting the presence of an IEC-converting gene (Beta-C-Phi-s); on the contrary, the *sea* and *sep* genes were not detected in any samples. The *sak* gene was found in all samples except for the two *S. aureus* ST59 samples, while the *chp* gene was absent only in the *S. aureus* ST10 samples. The predominant IEC variant [10] was type B (*sak*, *chp*, and *scn*), followed by type C (*chp*, *scn*), and type E (*sak*, and *scn*).

### 2.8. Phylogenetic Analysis

Single nucleotide polymorphisms (SNPs) were taken from 171 informative sites in the core genome within the entire dataset. A maximum-likelihood phylogenetic tree from this matrix was built, which resolved three major clades and three single leaves that agree with species-wide phylogeny (Figure 1). The largest clade encompassed all the *S. aureus* strains, irrespective of the isolation source (dog or human), which were further grouped according to the ST. A second clade grouped all the *S. pseudointermedius* strains; finally, a small clade included all the *S. epidermidis* strains.

To better investigate the intraspecies relationships, the SNP matrixes were re-calculated for those clusters that exhibited multi-host or possible host-switching events only, i.e., the *S. aureus* and *S. epidermidis* clusters, while the *S. pseudointermedius* cluster was considered as the baseline for a species recurrently isolated from a single-host (Appendix A). The distance matrix for the *S. aureus* isolates only, reporting the total number of different SNPs between any pair of isolates, showed a distance of about five SNPs per year with some deviations depending on the specific ST. Indeed, the SNP distance reached the lower value in the case of the S. aureus ST59 and ST10 strains that displayed a minimum value of three and two SNPs/year, respectively. On the contrary, the isolates belonging to ST8 and ST22 showed a minimum value of 4 and 11 SNPs/year, respectively.

A possible host-switching event was observed within the *S. aureus* clade where the closest related strains (Sau_17_DO_ST8_123220) and (SAU_17_FA_ST8_123232), only differing by four core genome SNPs, belonged to the dog and the father, respectively (Appendix A). Assuming a mutation rate of 8 SNPs per year [11] for *S. aureus* ST8, we suggest about 6 months of divergence for the ST8 strains.

Interestingly, the SNP variation analysis revealed a different genomic stability between *S. aureus* and *S. pseudintermedius* over time. In particular, within the *S. aureus* clusters, the ST8, ST10, and ST59 displayed lower genomic differences in terms of SNPs variation. Conversely, the ST22 strains were less genomically similar, as they showed the highest SNP differences per year (i.e., 11 SNPs). Surprisingly, the *S. aureus* ST707 and the ST7204 strains SNP distance was one of the lowest (i.e., 5 SNPs) found within the investigated dataset, despite belonging to different STs.

In regard to the *S. pseudintermedius* strains, the ST2169 can be compared to *S.aureus* ST22 strains in terms of genomic stability as a difference of 10 SNPs was found among them. The highest differences in terms of SNPs can be seen among ST2168 strains, with a difference of about 21 SNPs per year.

## 3. Discussion

Of the 25 *Staphylococcus* isolates collected in the present study, all but the four *S. pseudintermedius* strains were of a typically human origin [12,13,14], even though the *S. aureus* strains were isolated from both humans and the family dog. On the contrary, *S. pseudintermedius*, typically colonizing dogs’ skin [13], was isolated during the entire sampling time span from the dog only. This suggests the possibility for the unidirectional transfer of *S. aureus* circulating in the human reservoir from humans to dogs and not the other way around.

*S. aureus* ST8 that colonized the mother, the father, and the dog in 2017 and the grandparents in 2018, is known to be a pandemic clonal lineage of hypervirulent, community-acquired, methicillin-resistant *S. aureus* that includes the well-known PFGE strain USA300 [14]. This ST is the predominant MRSA isolated in North America and some parts of Europe [15] and it is known to be multiresistant and mecA-positive. Notably, this ST was spread both in symptomatic (father, grandmother, and grandfather) and asymptomatic (mother) humans. This poses serious concerns regarding (i) the possibility for such a virulent ST to provoke folliculitis, (ii) the possibility for asymptomatic people to spread this ST to immunocompromised people, and (iii) the possibility for this ST to be transferred from humans to pets and vice-versa. Since conserved prophage and plasmid replicon patterns were shared by Sau_17_MO_ST8_123221 Sau_17_DO_ST8_123220 Sau_18_GM_ST8_153686-6, it is possible to speculate a transfer from the mother to the dog.

The *S. aureus* ST59, isolated from the grandparents in two different sampling sessions (2018 and 2019), is the most prevalent sequence type isolated in Eastern Asia, Europe, and North America, representing the major cause of skin and soft tissue infections, as well as being described as a commensal colonizer [16,17]. Thus, we cannot exclude that the severe symptoms displayed by both grandparents could be ascribable to ST8 and/or ST59. In addition, in our dataset, this ST, as well as ST8, resulted in being multiresistant, with particular reference to beta-lactamases, tetracycline, methicillin, aminoglycosides macrolides, erythromycin, and streptogramin B.

It is important to highlight that the SAU_18_CH_ST707_153686-4 and SAU_18_DO_ST7204_153686-1 strains, isolated from the child and the dog, respectively, despite belonging to different STs (707 and 7204), were very close in terms of SNPs and shared the same prophage profiles, antibiotic resistance genes, and plasmid profile, and were the only two strains displaying a common mobilome. This, together with the remark that ST7204 was isolated for the first time in this study from a dog, led us to hypothesize that this strain could have been transferred to the dog via the child.

In addition, this study revealed the same IEC type in both dog and human strains, thus suggesting the possibility that all the *S. aureus* found in the dog had a human source. IEC genes are involved in host immune evasion and provide *S. aureus* with a unique mechanism to adapt to and counteract the human host [18].

Although ubiquitous on human skin, *S. epidermidis* was only isolated from the child’s mouth in 2017 and from the grandparents’ noses in 2019. The three isolates belonged to different STs, and this suggests their independent origin. Even though *S. epidermidis* is one of the major representative taxa of human skin microbiota, it is considered an opportunistic pathogen and has been found to be a carrier and reservoir for antibiotic-resistance genes, particularly those that do not impose a major fitness cost on the bacteria, such as methicillin resistance coding elements [19,20]. Accordingly, there is evidence suggesting the possibility of the transfer of methicillin resistance cassettes from *S. epidermidis* to *S. aureus* [21]. The three isolates characterized herein shared the presence of aminoglycosides, Fosfomycin, methicillin, macrolides, erythromycin, and streptogramin A and B resistance genes. While *S. epidermidis* infections rarely develop into severe diseases, their frequency, and the fact that they are multiresitent and thus extremely difficult to treat, represent a serious burden for public health.

Taking into consideration the SNP distances, we have drawn a hypothetical transmission path for the *S. aureus* strains that has been summarized in Figure 2. Within the investigated family from 2017 and 2019, five different S. aureus STs had circulated. The *S. aureus* ST8 strains circulated in 2017 and 2018 between the family members, excluding the child. The three ST8 strains isolated in 2017 differed for 4–10 SNPs; the two ST8 strains isolated from the grandparents in 2018 differed by 6 and 14 SNPs from the closest 2017 strains and just 8 SNPs were among them. From a conservative point of view, this suggests a unidirectional transfer from grandfather to grandmother (Figure 2).

Conversely, in the case of the *S. aureus* ST10, the circulation might have originated from the mother in 2018, where the isolates persisted with a number of SNPs equal to three and then it was transferred to the child. The *S. aureus* ST22 originated from the mother in 2017 and then moved to the father in 2018 where it persisted until 2019 with a slightly higher SNP distance (11 SNPs). The *S. aureus* ST59 was probably transferred to the grandfather from the grandmother after a divergence of three SNPs. Finally, the *S. aureus* ST707-ST2704 circulated between the child and the dog, and the two strains remained closely related as they differed by only five SNPs.

This evidence, together with the presence of the same antibiotic resistance gene pattern, EIC type variant, and plasmid and prophage patterns provide the proof for the anthroponotic transmission of *S. aureus* (ST707-ST2704) from the child to the family dog and suggest that companion animals can potentially have a role in the spread of *Staphylococcus* between humans and animals, thus posing a serious concern about MRSA spreading within human and animal microbial communities.

## 4. Materials and Methods

### 4.1. Study Design

A single family, composed of father, mother, grandfather, grandmother, a child, and a dog, was yearly sampled in the period 2017–2019. Sterile test tubes together with swabs (Modified Amies Medium, Meus s.r.l. Piove di Sacco, Italy) were used to collect samples from human skin lesions, intact skin, and mucous membranes (hands, mouth, and nostrils) in symptomatic subjects and from mucous membranes (hands, mouth, and nostrils) in asymptomatic ones. In dogs, mouth, ear canals, axillary region, and groin were sampled in the same way. Swabs were stored in refrigerated boxes at 6–8 °C and were cultured within 24/36 h of collection.

The dog was monitored as the suspected cause of the infection by the attending physician, but it never showed skin infections. The study was ethically approved by IZSVe ethical committee (opinion n. CE_IZSVE 02/2022).

### 4.2. Microbiological Investigations

Bacterial isolation was performed according to standard laboratory culture techniques. Briefly, each swab was inoculated onto the following solid media: nutrient blood agar (Blood Agar Base, Biolife, Milano, Italy; supplemented with 5% defibrinated sheep blood, Allevamento Blood, Teramo, Italy), Enterobacteriaceae selective medium (McConkey agar, Oxoid, Basingstoke, UK), methicillin-resistant Staphylococci selective medium, (CHROMagar^®^ MRSA II, BD BBL™, Heidelberg, Germany), and nutrient broth (Mueller–Hinton broth, Biokar Diagnostics, Alonne, FR, enriched with 6.5% sodium chloride, Sigma-Aldrich s.r.l., Milano, Italy), and then incubated at 37 °C in aerobic conditions for 24–48 h.

In case of the absence of microbial growth on the plates after 24 h of incubation, in the presence of turbidity of the broth, seeding onto methicillin-resistant Staphylococci selective medium was carried out as described above.

Staphylococcal colonies were recognized on nutritive medium, according to colony morphology, Gram stain appearance and cellular morphology, and catalase and coagulase tube tests. Suspected pink to mauve colonies grown on the methicillin-resistant Staphylococci selective medium were suspected and confirmed as being MRSA with a multiplex-PCR targeting nuc and mecA genes [22]. S. aureus DSMZ 11729 was used as positive control.

Species identification was performed by MALDI-TOF MS: Microflex LT instrument (MALDI Biotyper, Bruker Daltonics) equipped with FlexControl software (version 3.3, Bruker Daltonics).

### 4.3. DNA Extraction and Whole-Genome Sequencing

Genomic DNA was extracted from single colonies cultured on Blood Agar plates, using the QIAamp DNA Mini kit (QIAGEN), according to the manufacturer’s protocol, and quantified with a Qubit 3.0 Fluorometer (Life Technologies). Libraries for whole genome sequencing were prepared starting from 1 ng of genomic DNA with the Nextera XT DNA sample preparation kit (Illumina). Paired-end high-throughput sequencing (2 × 251 bp) was performed on a MiSeq sequencing platform, using the v3 chemistry.

### 4.4. Genome Assembly and Annotation

De novo contigs were generated for each genome using SPAdes (version 3.13) via parameters for 2 × 251 Illumina reads [23]. Contigs longer than 200 bp were retained. Chromosome assemblies were annotated by Prokka (version 1.11) [24]. The genus and species were confirmed by using the online version of kmerFinder tool (version 3.2) [25]. The multi-locus sequence type (MLST) was derived from the assemblies using the MLST web application (version 2.0) [26]. Antibiotic resistance genes were identified using the ResFinder web application (version 4.1) [27] accessed in March 2020. Finally, integrated prophages were searched using the PHASTER web server (accessed in March 2020) [28], while potential plasmidic contigs were checked by PlasmidFinder web application (version 2.1) [29]. Immune evasion cluster (IEC) genes sequences were obtained from Genebank using accession numbers reported in the work of Ahmadrajabiet and colleagues [10]. IEC genes sequences, at protein level, were searched by similarity using BLAST (https://blast.ncbi.nlm.nih.gov/Blast.cgi accessed on 20 November 2021) against annotated protein coding genes returned by Prokka.

### 4.5. Accession Number(s)

The genome sequence data from this study have been uploaded to Genebank under the bioproject accession number PRJNA762216.

### 4.6. Phylogenetic Analysis

The phylogenetic analysis was conducted on the alignments of the core genome SNP matrix as returned by KSNP3 software (version 3.91) [30], using a kmer value equal to 21. The alignments were imported into MEGA X phylogenetic package [31] to derive SNP distance among isolates; further, a Maximum Likelihood phylogenetic tree was calculated using GTR gamma-4 substitution model.

## Figures and Tables

**Figure 1 pathogens-11-00802-f001:**
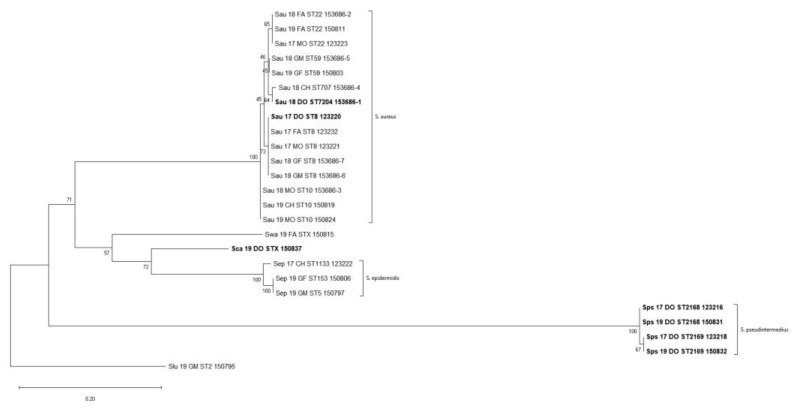
Whole Dataset SNP-based Phylogenetic Tree. The evolutionary history was inferred by using the Maximum Likelihood method and General Time Reversible model. The tree with the highest log likelihood (−1000.76) is shown. The percentage of trees in which the associated taxa clustered together is shown next to the branches. Initial tree(s) for the heuristic search were obtained automatically by applying Neighbor-Join and BioNJ algorithms to a matrix of pairwise distances estimated using the Maximum Composite Likelihood (MCL) approach, and then selecting the topology with superior log likelihood value. There was a total of 171 positions in the final dataset. Samples isolated from the dog are reported in bold.

**Figure 2 pathogens-11-00802-f002:**
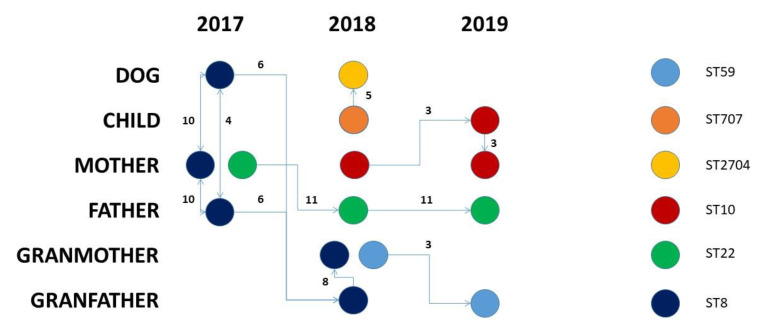
Evolutionary hypothesis. The figure shows the evolutionary hypothesis regarding the transmission of S. aureus strains among the investigated subjects over time. Links among samples are placed based on allelic distances, while arrows are placed according to the sampling date when possible.

**Table 1 pathogens-11-00802-t001:** Dataset. The table reports the strains involved in this study, the species, the date of collection, and the source of isolation in terms of family member and body district. Seven Loci Sequence Type was derived from the assemblies (see Appendix A for extended results). ND: Not Determined (Schema not available).

Sample	Organism	Collection Date	Family Member, District	ST
Sau_17_FA_ST8_123232	*S. aureus*	2017	father, skin	ST8
Sau_17_MO_ST8_123221	*S. aureus*	2017	mother, mouth	ST8
Sau_17_DO_ST8_123220	*S. aureus*	2017	dog, armpit	ST8
Sau_18_GF_ST8_153686-7	*S. aureus*	2018	grandfather, nose	ST8
Sau_18_GM_ST8_153686-6	*S. aureus*	2018	grandmother, armpit	ST8
Sau_18_MO_ST10_153686-3	*S. aureus*	2018	mother, nose	ST10
Sau_19_MO_ST10_150824	*S. aureus*	2019	mother, nose	ST10
Sau_19_CH_ST10_150819	*S. aureus*	2019	child, nose	ST10
Sau_17_MO_ST22_123223	*S. aureus*	2017	mother, nose	ST22
Sau_18_FA_ST22_153686-2	*S. aureus*	2018	father, nose	ST22
Sau_19_FA_ST22_150811	*S. aureus*	2019	father, nose	ST22
Sau_18_GM_ST59_153686-5	*S. aureus*	2018	grandmother, nose	ST59
Sau_19_GF_ST59_150803	*S. aureus*	2019	grandfather, nose	ST59
Sau_18_CH_ST707_153686-4	*S. aureus*	2018	child, nose	ST707
Sau_18_DO_ST7204_153686-1	*S. aureus*	2018	dog, mouth	ST7204
Sca_19_DO_STX_150837	*S. capitis*	2019	dog, armpit	ND
Sep_19_GF_ST153_150806	*S. epidermidis*	2019	grandfather, nose	ST153
Sep_17_CH_ST1133_123222	*S. epidermidis*	2017	child, mouth	ST1133
Sep_19_GM_ST5_150797	*S. epidermidis*	2019	grandmother, nose	ST5
Slu_19_GM_ST2_150795	*S. lugdunensis*	2019	grandmother, nose	ST2
Sps_17_DO_ST2169_123218	*S. pseudintermedius*	2017	dog, mouth	ST2169
Sps_19_DO_ST2169_150832	*S. pseudintermedius*	2019	dog, foreskin	ST2169
Sps_17_DO_ST2168_123216	*S. pseudintermedius*	2017	dog, mouth	ST2168
Sps_19_DO_ST2168_150831	*S. pseudintermedius*	2019	dog, mouth	ST2168
Swa_19_FA_STX_150815	*S. warneri*	2019	father, nose	ND

**Table 2 pathogens-11-00802-t002:** Antibiotic-Resistance Genes. The table reports antibiotic-resistance genes as detected by the ResFinder webserver.

Sample	blaZ	ant(6)-Ia	aph(3)-III	aadD	aac(6)-aph(2)	cat(pC221)	fosB	fusB	mecA	mph(C)	msr(A)	erm(C)	erm(B)	vga(A)	vga(A)LC
Sau_17_FA_ST8_123232	x	x	x						x	x	x				
Sau_17_MO_ST8_123221	x	x	x						x	x	x				
Sau_17_DO_ST8_123220	x	x	x						x	x	x				
Sau_18_GF_ST8_153686-7	x	x	x						x	x	x				
Sau_18_GM_ST8_153686-6	x	x	x						x	x	x				
Sau_18_MO_ST10_153686-3	x														
Sau_19_MO_ST10_150824	x									x	x				
Sau_19_CH_ST10_150819	x														
Sau_17_MO_ST22_123223	x														
Sau_18_FA_ST22_153686-2	x														
Sau_19_FA_ST22_150811	x														
Sau_18_GM_ST59_153686-5	x														
Sau_19_GF_ST59_150803	x														
Sau_18_CH_ST707_153686-4	x														
Sau_18_DO_ST7204_153686-1	x														
Sca_19_DO_STX_150837	x														
Sep_19_GF_ST153_150806	x				x	x	x								
Sep_17_CH_ST1133_123222	x			x			x	x		x	x				
Sep_19_GM_ST5_150797	x			x	x	x	x	x	x			x		x	x
Slu_19_GM_ST2_150795						x									
Sps_17_DO_ST2169_123218	x	x	x										x		
Sps_19_DO_ST2169_150832	x	x	x										x		
Sps_17_DO_ST2168_123216	x	x	x			x							x		
Sps_19_DO_ST2168_150831	x	x	x			x							x		
Swa_19_FA_STX_150815															

**Table 3 pathogens-11-00802-t003:** Intact Prophage Sequences. The table reports the intact prophage sequences only as predicted by PHASTER web server. For brevity, the “Staphy” prefix was removed from the prophage names.

S. aureus	69	11	phi2958PVL	StauST398_2	phiJB	phi2958PVL	P282	YMC/09/04/R1988
Sau_18_MO_ST10_153686-3	x			x				
Sau_19_MO_ST10_150824		x	x					
Sau_19_CH_ST10_150819	x		x					
Sau_18_CH_ST707_153686-4					x			
Sau_18_DO_ST7204_153686-1					x			
Sau_17_FA_ST8_123232								
Sau_17_MO_ST8_123221							x	x
Sau_17_DO_ST8_123220							x	x
Sau_18_GM_ST8_153686-6							x	x
Sau_18_GF_ST8_153686-7						x	x	
** *S. epidermidis* **	**StB20_like**	**Ipla5**						
Sep_17_CH_ST1133_123222								
Sep_19_GF_ST153_150806	x	x						
Sep_19_GM_ST5_150797								

**Table 4 pathogens-11-00802-t004:** Plasmid content. The table reports the plasmid sequences as detected by the plasmid finder web server.

Sample	rep19	rep39	rep7a	repUS43	rep7c	rep16	repUS5	rep20	repUS35	rep22	repUS19	repUS12	rep5b
Sau_17_DO_ST8_123220	x				x	x							
Sau_17_FA_ST8_123232	x		x		x	x							
Sau_18_GF_ST8_153686-7	x				x	x							
Sau_18_GM_ST8_153686-6	x				x	x							
Sau_17_MO_ST8_123221	x				x	x							
Sau_18_CH_ST707_153686-4						x	x						
Sau_18_DO_ST7204_153686-1						x	x						
Sau_18_FA_ST22_153686-2													
Sau_17_MO_ST22_123223													
Sau_19_FA_ST22_150811													
Sau_19_MO_ST10_150824						x	x						
Sau_18_MO_ST10_153686-3													
Sau_19_CH_ST10_150819													
Sau_19_GF_ST59_150803													
Sau_18_GM_ST59_153686-5													
Sca_19_DO_STX_150837								x					
Sep_17_CH_ST1133_123222		x	x							x			
Sep_19_GF_ST153_150806			x										
Sep_19_GM_ST5_150797			x								x	x	x
Slu_19_GM_ST2_150795			x										
Sps_17_DO_ST2169_123218				x									
Sps_17_DO_ST2168_123216			x										
Sps_19_DO_ST2169_150832				x									
Sps_19_DO_ST2168_150831			x										
Swa_19_FA_STX_150815									x				

**Table 5 pathogens-11-00802-t005:** IEC Genes. The table reports the presence of IEC Genes, including the IEC variant.

Sample	scn	sep	sea	sak	chp	IEC-Variant	Sample
Sau_17_DO_ST8_123220	x			x	x	B	Sau_17_DO_ST8_123220
Sau_17_FA_ST8_123232	x			x	x	B	Sau_17_FA_ST8_123232
Sau_18_GF_ST8_153686-7	x			x	x	B	Sau_18_GF_ST8_153686-7
Sau_18_GM_ST8_153686-6	x			x	x	B	Sau_18_GM_ST8_153686-6
Sau_17_MO_ST8_123221	x			x	x	B	Sau_17_MO_ST8_123221
Sau_18_CH_ST707_153686-4	x			x	x	B	Sau_18_CH_ST707_153686-4
Sau_18_DO_ST7204_153686-1	x			x	x	B	Sau_18_DO_ST7204_153686-1
Sau_18_FA_ST22_153686-2	x			x	x	B	Sau_18_FA_ST22_153686-2
Sau_17_MO_ST22_123223	x			x	x	B	Sau_17_MO_ST22_123223
Sau_19_FA_ST22_150811	x			x	x	B	Sau_19_FA_ST22_150811
Sau_19_MO_ST10_150824	x			x		E	Sau_19_MO_ST10_150824
Sau_18_MO_ST10_153686-3	x			x		E	Sau_18_MO_ST10_153686-3
Sau_19_CH_ST10_150819	x			x		E	Sau_19_CH_ST10_150819
Sau_19_GF_ST59_150803	x				x	C	Sau_19_GF_ST59_150803
Sau_18_GM_ST59_153686-5	x				x	C	Sau_18_GM_ST59_153686-5

## Data Availability

The genome sequence data from this study have been uploaded to Genebank under the bioproject accession number PRJNA762216.

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
