# Peer review of "Anthroponotic-Based Transfer of Staphylococcus to Dog: A Case Study"

_pathogens, 2022, doi:10.3390/pathogens11070802_

Round 1
Reviewer 1 Report
In this paper the authors study the transmission of staphylococcus ssp. within a family including a dog. Overall study design and execution appears sound and results support the conclusions.
Data show extensive transmission between humans consisting of multiple STs and primarily S. aureus species. There is evidence of one transmission from the child to the dog. This is certainly of interest although it is not the first time anthroponotic transmission of S. aureus has been reported. Transmission of S. aureus to pets can be of significance for public health especially if pets can then pass the bacterial on to other humans.
The authors report that there were three symptomatic cases with USA300 in this family outbreak, but no information is provided on the symptoms and the resolution of the infection. It is important to privide this information in the manuscript.
Of note, despite this ST8 strain going through all the human family members, it wasnt this strain that managed to transfer to the dog- suggesting that this is a rare event.
Author Response
We wish to thank the reviewer for having handled the manuscript.
We added a new paragraph regarding the Clinical context of the study in the Results section starting from information already present in the Materials and Methods section.
Moreover, we modified the Study design paragraph in MM section accordingly.
Best regards,
Massimiliano Orsini and Carmen Losasso

Reviewer 2 Report
The manuscript describes an interesting study regarding the possible transmission of pathogens between humans and their house pets. However, it would have been interesting to study several households and thier pets instead of focusing only on one household, but i can imagine that it might be complicated to enroll more households for several years.
Author Response
We wish to thank the reviewer for having handled our manuscript.
We fully agree with the reviewer that adding more family outbreacks could be of great value to the manuscript however it is complicated to find the clical cases and enroll them for several years.